# The Central Anatomical Question: Treatment of Lymphoma Within Border-Zone Anatomical Sites Adjacent to the Central Nervous System

**DOI:** 10.3390/cancers17203392

**Published:** 2025-10-21

**Authors:** Candace Marsters, Chai Phua, Maria MacDonald, Gabriel Boldt, Seth Climans

**Affiliations:** 1Department of Oncology, University of Alberta, Edmonton, AB T6G 1Z2, Canada; marsters@ualberta.ca; 2Division of Hematology, Department of Medicine, Western University, London, ON N6A 5W9, Canada; 3Department of Oncology, Western University, London, ON N6A 5W9, Canada; 4Department of Clinical Neurosciences, Western University, London, ON N6A 5A5, Canada; 5Verspeeten Family Cancer Centre, London Health Sciences Centre, London, ON N6A 5W9, Canada

**Keywords:** lymphoma, B-cell lymphoma, central nervous system, cavernous sinus, circumventricular organs, cranial nerves, pituitary gland, dura

## Abstract

Understanding where lymphoma in the central nervous system begins and spreads is important for both treatment and prognosis. Standard definitions of the central nervous system, which often focus on the brain and spinal cord, usually leave out certain “border-zone” areas that can have important implications when deciding on treatment regimens. In this review, we explore these challenging regions and discuss how lymphoma involvement in these areas should be classified to guide treatment planning. By better defining these boundaries, our goal is to support clearer treatment decisions to help improve patient care.

## 1. Introduction

Lymphomas of the central nervous system (CNS) are grouped into primary CNS lymphoma (PCNSL) and secondary CNS lymphoma (SCNSL; systemic lymphoma and synchronous CNS involvement at diagnosis or CNS involvement in disease progression/relapse). CNS involvement of non-indolent lymphomas generally has a worse prognosis when compared to lymphomatous disease not involving the CNS [1,2]. The gold standard for diagnosis of CNS involvement continues to be tissue biopsy, while histopathological analysis through examination of CSF with cytology, molecular and gene rearrangement analysis, ctDNA analysis, flow cytometry, and biochemical analysis also assists in diagnostic evaluation [2,3]. Furthermore, imaging using magnetic resonance imaging (MRI) with contrast enhancement and positron emission tomography (PET) can be used to help identify the presence and location of CNS lesions, but should be performed using the current recommended imaging consensus guidelines [4].

The current generally agreed upon boundaries of the CNS are often broadly categorized to include the cerebrum, basal ganglia, thalamus, interbrain, ventral hypothalamus, brainstem, and spinal cord [5]. Although this working definition offers a basic set of CNS structures, it fails to capture the nuances of these and the surrounding structures. One such key player is the blood–brain barrier. The blood–brain barrier consists of a highly selective barrier made up of endothelial cells, mural cells, the basement membrane matrix, and a variety of immune cells [6]. It regulates the complex exchange of molecules, ions, and cells between the peripheral bloodstream and CNS using a diverse range of physiological processes. Aside from the blood–brain barrier, there are other pathways that enable circulating substances to contact the CNS, such as the circumventricular organs (CVOs; including the choroid plexus, pineal gland, median eminence, area postrema, subfornical organ, subcommissural organ, and organum vasculosum of the lamina terminalis), the meninges, nerve ganglia, and the nucleus tractus solitarius [7]. Other barriers include the blood–retinal barrier and the blood–cerebrospinal fluid (CSF) barrier [8,9]. Importantly, new discoveries have elevated our understanding of how the brain orchestrates immunological privilege, unveiling a diverse and complex interplay at these brain borders. As an example, dural lymphomas are considered outside the CNS and are successfully treated with non-CNS-penetrating therapies, but this area may be at higher risk for seeding disease into the CNS. An important discovery regarding CSF drainage has shown that the dura is, in fact, a relevant part of this drainage system. Previously, CSF was thought to be drained entirely through the cerebral venous system, but new research has revealed that CSF drainage occurs through the dura-meningeal lymphatic system, which flows into cervical lymph nodes rather than draining through the cerebral venous system [10]. This new information opens up novel possible pathways for systemic immune cell entry and invasion into the CNS through connections of the traditional lymphatic system to the CNS-specific waste clearance system known as the glymphatic system. These nuances in the anatomy of the blood–brain barrier and lymphatic system along the borders of the CNS have important context regarding systemic lymphoma spread into the CNS, with implications for tumor cell origin, CNS tropism, and immune-privileged lymphoma subtypes [11,12,13]. It also influences the routes and limitations for therapy regimens and drug delivery options of lymphoma treatments.

Lymphoma lesions within the brain parenchyma, optic tract, olfactory nerve, spinal cord, CSF space, and the leptomeninges are commonly considered consistent with CNS involvement and CNS-directed lymphoma treatment is recommended for these areas. CNS-directed therapies for B-cell lymphomas require consideration for specific blood–brain barrier-penetrating drug types (chemotherapies, BTK inhibitors, CAR T-cell therapy), alternative drug dosing (such as high-dose intravenous methotrexate), or use of intrathecal methotrexate. These CNS-specific regimens can carry a higher risk of toxicity and require additional clinical fitness assessments and close monitoring for therapy tolerability [14]. Ambiguities arise with whether to define lymphoma invasion in locations of the cavernous sinus, dura, circumventricular organs, pituitary gland, cranial nerves, and specific locations within the eye as representing actual CNS involvement. This becomes important for treatment considerations, particularly when there is invasion of these areas without involvement of more conventional CNS regions. Current guidelines lack nuanced recommendations for treating lymphoma isolated within these areas. In this article, we seek to better understand the boundaries of the central nervous system anatomy relating to lymphoma location. A consensus definition of whether or not to treat these border-zone anatomical sites (Figure 1) with therapies that cover the CNS will assist clinicians in prognosis and treatment planning. Though certain anatomical structures of the eye are considered connected to the CNS, recommendations regarding structures of the eye are beyond the scope of this article.

## 2. Anatomical Sites of Interest

### 2.1. The Dura

The dura makes up the outer portion of the meningeal layer that surrounds the brain parenchyma, overlaying the inner leptomeningeal layer, which is made up of the arachnoid and pia mater. Recent discoveries have highlighted important anatomical distinctions between the dura and the leptomeningeal layer, as its vasculature lacks tight junctions but contains fenestrations that allow immunological surveillance and the exchange of material between the peripheral circulation and that of the CNS. It contains both resident immune cells and an active lymphatic system, where CSF drainage can occur through the subarachnoid spaces directly into the dural lymphatic system, with drainage even being seen along the dura of cranial nerves and nerve sheaths [15,16,17,18,19]. The dural lymphatic system was previously thought to drain directly into the CNS venous system, but new evidence suggests drainage into cervical lymph nodes [10]. The wide array of connections between the dura and CSF offers multiple routes for CNS invasion of malignant cells. The dura also contains important connections between the skull bone marrow with evidence of direct trafficking of specialized immune cells from the skull bone marrow to the dura, presenting an alternative access route for malignant cell access to the CNS aside from hematogenous spread. Once malignant cells gain access to these sites, a process of immunomodulation within these pivotal immunosurveillance areas may occur with potential to alter barrier function [10,19,20]. The dura is considered outside the blood–brain barrier, but due to its location and interconnection to the CSF space, it requires added deliberation when considering lymphoma treatment for disease in this area.

Clinical exam findings for lesions in the dura can present as headaches, seizures, cranial nerve deficits, visual disturbances, or other focal neurological deficits. MRI of dural lesions typically reports extra-axial masses often with a dural tail that diffusely enhances. These are most commonly seen at the cerebral convexities and may be associated with underlying parenchymal vasogenic edema or bone erosion [21]. Small observational studies and systematic case review of primary isolated dural lymphoma lesions suggest they have a better prognosis when compared to PCNSL. As well, cases using CNS-directed treatment did not provide a survival benefit compared to standard non-CNS-directed therapies for isolated dural lesions, making it reasonable to proceed with systemic-only therapies in isolated dural lesions without evidence of lymphomatous invasion into meninges, CSF, or brain parenchyma [22,23].

**Expert Opinion:** In cases of primary dural lesions or systemic lymphoma spread to the dura and in the absence of dissemination into meninges, CSF, or brain parenchyma, expert recommendation is to treat the disease in this area as a non-CNS structure. Dural involvement is located outside the blood–brain barrier; therefore, systemic-only therapies with limited or no CNS penetration can treat this area (e.g., R-CHOP). However, it is crucial to ensure that the disease is indeed isolated to the dura. If there is evidence of deeper involvement in the CNS, the addition of CNS-directed treatment therapy should be considered. Given the location directly adjacent to the leptomeningeal layer, the risk of dissemination of systemic immune cells into the CSF space, additional investigations should be considered. We recommend an initial CSF cytology and flow cytometry assessment. Furthermore, CNS imaging with PET and MRI Brain with gadolinium should be completed prior to the onset of treatment and at the end of treatment. Repeat CSF can be considered only if clinical concern arises for CNS invasion or in individuals with systemic lymphoma and high-risk features for CNS invasion.

### 2.2. The Cavernous Sinus

The cavernous sinuses are located on each side of the sphenoid sinus, sella, and pituitary gland and extend from the superior orbital fissure in front to the temporal bone. They are in close proximity to the brain, eye, nasopharynx, and ear and contain both major arterial and venous structures that supply the brain and meninges, which can seed lymphoma into these areas. As well, there are multiple cranial nerves that convene in this area which can be affected by direct invasion resulting in neurolymphomatosis [24,25]. The venous portion is considered extradural in location and considered outside the blood–brain barrier, while the cranial nerves run within the intradural space at this location. As with the dura, new evidence suggests that a meningeal lymphatic system is also located in the cavernous sinus and is involved in CSF drainage [26]. Lymphomas can invade the cavernous sinus either by direct extension from adjacent structures, such as the marrow of surrounding bones or the nasopharynx, or by hematogenous dissemination. Given the location of multiple cranial nerves within a small space, there are multiple physical exam findings that can present with lymphomatous invasion of this area [25]. A general term for symptoms relating to the cavernous sinus is “cavernous sinus syndrome” and includes headache, ophthalmoplegia, diplopia, facial sensory loss, Horner’s syndrome, chemosis, and proptosis [25,27]. Imaging of lymphoma within the cavernous sinus is best performed with MRI given the superior soft tissue contrast resolution. Lymphoma will often present with diffusion restriction owing to the high cellularity of the tumor. Lesions generally demonstrate strong homogenous gadolinium enhancement best seen in post-contrast fat-suppressed T1W imaging [25,27,28]. Prognosis is variable depending on the extent of disease and which cavernous sinus structures are involved.

**Expert Opinion:** Theoretically, the venous structure that makes up the cavernous sinus is located outside the blood–brain barrier; however, given its critical location adjacent to and involving multiple brain structures, we suggest an aggressive approach to treatment when considering curative intent. For this reason, we suggest that most cases of cavernous sinus involvement should include CNS-directed therapy due to the proximity to adjacent structures deemed behind the blood–brain barrier. Of note, in patients who are not candidates for the addition of CNS-directed treatment regimens, sole involvement of the cavernous sinus may portend a better prognosis than lymphoma disease involving the brain parenchyma, given the direct access of systemic-only treatments to this anatomical area.

### 2.3. The Choroid Plexus and Circumventricular Organs

The choroid plexus and CVOs are small, specialized midline structures localized around the third ventricle and fourth ventricle of the brain, creating a barrier between the blood vessels, brain, and CSF. They involve a unique and specialized transporter system that transports molecules across cell membranes with distinctive region-specific variations in blood–brain barrier signature genes that are different from the typical blood–brain barrier [9,29]. The choroid plexus is the most well-known of these structures and is involved in the production of CSF through filtering of blood from a fenestrated endothelial network, then secreting this fluid through specialized endothelial cells. It is situated within the ventricles of the brain but outside the pia mater portion of the leptomeninges, separated by a specialized epithelial layer that separates it from the CSF compartment [10]. There is evidence that systemic immune cells can cross this barrier during states of systemic inflammation, though it is unclear whether this can contribute to the metastatic spread of lymphoma into CSF spaces [30]. For this reason, access to the CSF and leptomeningeal spaces by malignant immune cells is theoretically possible. Clinical symptoms of tumor infiltrate into this area are often non-specific but can culminate in hydrocephalus, with ensuing headache, seizure, bilateral vision changes, incoordination, and decreased level of consciousness [31]. On MRI brain imaging, they will be seen as a well-circumscribed intraventricular mass in contact with the ventricular surface that will avidly enhance with contrast [32]. In theory, lymphoma isolated to one of these regions could be sensitive to chemotherapy that does not cross the blood–brain barrier. CVOs are a theoretically relevant anatomical site but have limited clinical value given the difficulty of establishing radiographic evidence of isolated involvement of these complex entities.

**Expert Opinion**: There is scant literature reporting lymphomatous invasion specifically into these structures but given their location abutting the brain parenchyma and suspected permeability of immune cells into the adjacent CSF space, expert recommendation would favor inclusion of CNS-directed therapy when these structures are involved or suspected.

### 2.4. The Pituitary

The pituitary is an endocrine gland that regulates homeostatic, metabolic, and reproductive processes within the body. In adulthood, there are two functional lobes derived from different embryonic tissues. The anterior pituitary lobe, derived from the oral ectoderm, is under the control of the hypothalamus and mainly functions in secreting hormones. It is not strictly considered a brain structure. The posterior pituitary gland, derived from the neural ectoderm and considered an anatomic extension of the hypothalamus, contains axons originating from the hypothalamus that travel through the pituitary stalk and terminate as neurosecretory granules in the posterior pituitary; thus, it is an extension of the brain parenchyma. The anterior pituitary receives most of its blood supply through the hypothalamic-hypophyseal portal system, which is directly connected to the hypothalamus at the median eminence via an interconnect portal capillary system. The posterior pituitary mainly receives its blood supply from branches off the internal carotid artery and contains a separate capillary system for hormone secretion. The pituitary contains a unique capillary system that is fenestrated, allowing rapid secretion of hormones from pituitary cells, with the capillary system considered to be outside the blood–brain barrier. For this reason, the pituitary homogenously enhances with contrast agents in both CT (Computed Tomography) and MRI, while non-contrast MRI is often variable and requires dedicated pituitary imaging sequences. Tumors of this area often further invade into the cavernous sinus or skull base due to their adjacent location. As well, the rich vascularity of the pituitary gland places a higher risk of metastatic hematogenous spread [33].

Lymphoma invasion into this area can be asymptomatic, but most commonly results in headaches, including retroorbital pain. Hormone dysregulation more commonly affects the anterior pituitary compared to the posterior pituitary, resulting in hypothyroidism and adrenal insufficiency, but the presence of diabetes insipidus can influence survival. Other presentations include visual changes, diplopia, facial sensation changes, or nausea with abnormalities depending on the location and extent of invasion. A study looking specifically at primary lymphoma in this area found that most patients were managed using surgery, radiotherapy, and CNS-directed chemotherapies. Untreated, lymphoma in this area tends to have poor survival. Due to the rarity of isolated primary pituitary lymphoma or isolated pituitary invasion in secondary lymphoma, there remains a paucity of evidence to conclusively guide treatment protocols for lymphoma isolated to this specific anatomical location [34,35].

**Expert Opinion**: The pituitary gland is a complex structure with portions located both outside and within the blood–brain barrier. It also contains a distinctively altered organization of the blood–brain barrier at this location to allow for increased permeability of molecules. Given its critical location adjacent to and involving the brain parenchyma, we suggest an aggressive approach to treatment and that cases of pituitary involvement should prompt the addition of CNS-directed therapy to ensure microscopic extension into the CNS is adequately treated. Importantly, though, given the unique anatomy of the pituitary and the altered blood–brain barrier in this area, systemic-only therapy may have improved penetration compared to the brain parenchyma and should be considered in cases where CNS-directed therapy cannot be used. As well, hormone replacement therapy may need to be a consideration with disease in this area.

### 2.5. The Cranial Nerves

The cranial nerves are composed of 13 paired nerves that provide sensory and motor innervation mainly to the head and neck. These nerves have portions within the CNS that extend through the dura and into the periphery.

Neurolymphomatosis, direct infiltration of lymphoma into nerves, is a heterogenous disease and can occur with any cranial nerve. Infiltration of a cranial nerve may allow access of disease into the meninges, adjacent cranial nerves, and brain parenchyma through proximity [36]. Symptoms of nerve infiltration are dependent on the specific nerve that is involved, resulting in various cranial nerve deficits. Investigations for detection of infiltrative disease commonly begin with MRI, with and without contrast, electrophysiological nerve testing, and lastly, nerve biopsy. In addition, CSF testing can suggest lymphoma infiltration of the leptomeninges but is commonly negative on initial testing [36,37]. A recent study has shown secondary neurolymphomatosis had a worse prognosis with a higher proportion of disease progression after treatment and death when compared to those with primary neurolymphomatosis. It is proposed that treatment-resistant cells residing in peripheral nerves may be protected from initial systemic chemotherapy resulting in subsequent secondary propagation of disease [37]. Treatment regimens are variable for neurolymphomatosis in general and often include CNS-directed therapies such as high-dose methotrexate, rituximab among other chemotherapies, and stem cell transplant with variable, and often poor, outcomes [38,39]. In one study with a cohort of 40 patients with neurolymphomatosis, 20% had cranial nerve involvement, but treatment and outcome were not reported based on disease location [38]. Whether there is a worse prognosis or a higher chance of brain parenchymal involvement occurring when cranial nerves are specifically affected in neurolymphomatosis is unclear. Of important note, paraneoplastic, inflammatory, or viral neuropathies should be considered as a differential which can mimic neurolymphomatosis [36].

**Expert Opinion**: Cranial nerves are considered both outside and within the boundaries of the CNS, travelling from the periphery through the blood–brain barrier to lie within the brain parenchyma. Involvement of nerve areas within the subarachnoid space is firmly considered to be within the CNS and should be treated as such. Involvement of the nerve portion outside this area is more nuanced, and systemic-only therapies can be considered. A caveat being that distinguishing the extent of nerve infiltration is difficult, and so treatment typically covers both possibilities unless there is clear evidence, using either biopsy or imaging, that nerve infiltration is distant to the CNS boundary.

## 3. Conclusions and Future Directions

We recommend careful consideration be undertaken when there is high-grade lymphoma involvement within the dura, cavernous sinus, pituitary gland, circumventricular organs, and cranial nerves, given the proximity to CNS structures. This is summarized in Table 1. Prophylactic intravenous high-dose methotrexate in individuals with high-risk systemic lymphoma, as prognosticated using the CNS International Prognostic Index, has not been shown to result in a clinically meaningful reduction in CNS progression and is generally not recommended, but these prognostic indices do not address these higher-risk anatomical areas specifically [40]. Isolated involvement of these areas is rare but can occur and should be reviewed on an individual basis. Instead, it may be that the probability of CNS involvement of secondary lymphomas requires an alternative risk score system that instead considers molecular factors, cells of origin, and anatomical location of disease, including proximity to the CNS or location within immunoprivileged sites. Here, we have made treatment recommendations for high-grade lymphoma involvement in selected important border-zone anatomical areas that border the CNS. Unfortunately, due to the rarity of isolated disease at these sites, there is a paucity of evidence guiding these treatments, and so recommendations are based on anatomical features of these areas and the likelihood of invasion into the adjacent CNS. Low-grade lymphomas may be monitored if asymptomatic and indolent; however, treatment principles as outlined here would still apply. Advanced imaging modalities such as multiparametric MRI, together with CSF analysis for circulating tumor DNA, molecular profiling (e.g., *MYD88*, *CD79B*), and proteomic mass-spectrometry approaches, hold promise for achieving greater diagnostic precision and more accurate compartmental differentiation between CNS and systemic lymphoma [4,41,42,43]. Overall, more research is needed to guide evidence-based therapies for the best treatment of these areas, but this article provides interim basic guidelines for treatment.

## Figures and Tables

**Figure 1 cancers-17-03392-f001:**
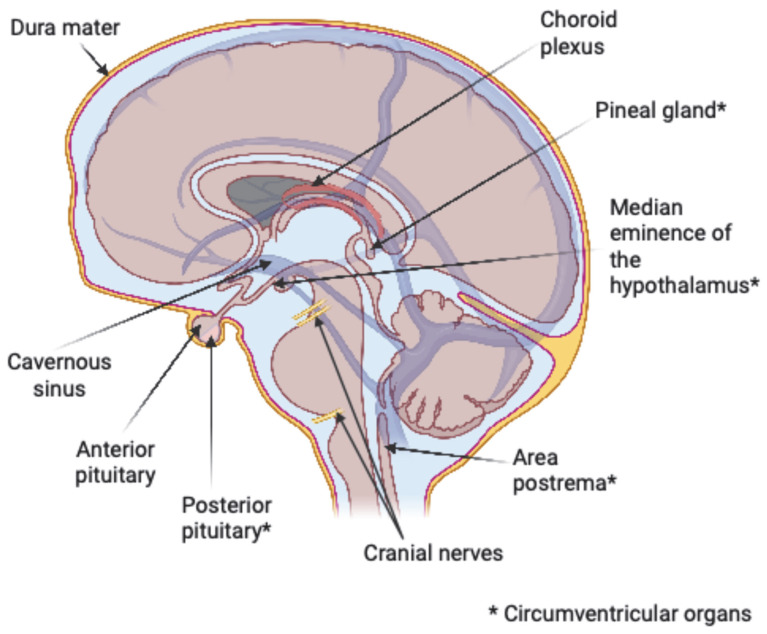
Anatomical location of the border-zone anatomical sites adjacent to the central nervous system of interest for treatment recommendations.

**Table 1 cancers-17-03392-t001:** Treatment recommendation for border-zone anatomical locations with high-grade lymphoma involvement.

	Anatomical Features	Common Symptoms	Treatment Considerations *
**Dura**	Adjacent to leptomeningeal layer. Has a role in CSF drainage into lymphatic system with direct access to systemic blood circulation, skull bone marrow, and cervical lymphatic system.	Headaches, seizures, cranial nerve deficits, visual disturbances, motor or sensory deficits.	CNS-directed therapy not required for isolated dural involvement, with close monitoring using CSF and imaging, assessing for meningeal or brain parenchyma invasion.
**Cavernous Sinus**	Adjacent to brain parenchyma while containing venous drainage from CNS, internal carotid artery, and multiple cranial nerves.	Headaches, cranial neuropathies (commonly ophthalmoplegia, facial numbness, Horner’s syndrome) exophthalmos, chemosis, vision changes, strokes.	Whenever possible, the inclusion of CNS-directed therapy to systemic therapy. This reflects the difficulty in precisely demarcating blood–brain barrier boundaries and the risk of microscopic extension beyond visible disease.
**CVOs**	Adjacent to brain parenchyma and CSF reservoirs with connections to systemic blood circulation.	Commonly asymptomatic but may present with symptoms consistent with increased intracranial pressure such as headache, blurred vision, or nausea.	
**Pituitary Gland**	Extension of/adjacent to brain parenchyma with systemically accessible fenestrated capillary vascular supply.	Symptoms secondary to hormonal dysregulation, diabetes insipidus, headaches, or visual changes.	
**Cranial Nerves**	Peripheral portion near the CNS with direct route into the brain parenchyma through the fascicle portion of the nerve.	Specific motor or sensory cranial nerve deficits which can include ptosis, diplopia, facial weakness, facial numbness, dysarthria, dysphagia, vertigo, or hearing loss.	CNS-directed therapy when affected portions of the nerve are directly adjacent to or within the dural boundary. Systemic therapy can be considered with only distal nerve involvement.

***** Expert opinion-based recommendations; CNS-directed therapy encompasses agents capable of penetrating the blood–brain barrier and may also include radiotherapy or surgical intervention when appropriate. Systemic therapy encompasses agents with limited or no CNS penetration, suitable for extracranial disease but not sufficient for CNS control. A multidisciplinary approach remains essential in managing these rare and complex cases.

## Data Availability

Not applicable.

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
