# Peer review of "The Central Anatomical Question: Treatment of Lymphoma Within Border-Zone Anatomical Sites Adjacent to the Central Nervous System"

_cancers, 2025, doi:10.3390/cancers17203392_

Round 1

Reviewer 1 Report

Comments and Suggestions for Authors

This is a timely and well-written review addressing an important and understudied clinical problem: how to classify and treat lymphomas involving anatomical “border-zones” between CNS and systemic compartments. The manuscript provides a thoughtful synthesis of anatomical, pathological, and therapeutic considerations and integrates expert opinion where evidence is scarce. The topic is highly relevant to clinicians managing primary and secondary CNS lymphoma.

  1. Some typos should be revised, such as use of both BBB and blood-brain barrier in introduction.
  2. Is there level of evidence that should be provided in Table 1?

Author Response

  1. Some typos should be revised, such as use of both BBB and blood-brain barrier in introduction.
    • We have re-reviewed the draft to fix typos and have standardized the language using blood-brain barrier throughout the manuscript
  2. Is there level of evidence that should be provided in Table 1?
    • Thank you for this suggestion allowing us to clarify the evidence level in Table 1. We have modified the treatment consideration portion and added a statement in lines 340-344. This was modified to better reflect the limited evidence base and the challenges in making prescriptive recommendations, given that much of this area remains “gray” but still allowing for a generalized approach for clinicians to refer to. 

Reviewer 2 Report

Comments and Suggestions for Authors

This manuscript discusses a rare but important issue about lymphoma, such as how it affects areas near the brain and nervous system that doesn’t fit precisely into either “inside the brain” or “outside the brain”. It highlights new findings about how fluid and immune cells move around the above border areas, which could change how clinicians consider treating the disease. The review is original because it efficiently discusses areas such as the dura, cavernous sinus, pituitary, and certain nerves, and it brings in recent discoveries about lymphatic drainage to explain why these brain regions matter. Overall, the article is clear, well-structured, and supported with references, though some treatment advice relies more on expert opinion than solid evidence. The Figures (1) and tables (1) are helpful but could be made clearer.

Suggestions:

  1. Remove redundancy in the introduction (such as BBB).
  2. Revise the sentence regarding CSF drainage for more clarity in introduction (line no. 73).
  3. Elaborate clearly the difference between treatments that directly target the brain and nervous system (“CNS-directed”) and those that only work on the rest of the body (“systemic-only”).
  4. Point out more clearly which statements are backed by research evidence, and which are based mainly on expert opinion.
  5. Expand the section on future directions to include the role of new imaging methods, genetic/molecular testing, and large patient registry studies.
  6. Include the treatment approaches in Figure 1.
Comments on the Quality of English Language

Only Minor changes required.

Author Response

  1. Remove redundancy in the introduction (such as BBB).
    • We have revised the manuscript to standardize the language using blood-brain barrier throughout.
  2. Revise the sentence regarding CSF drainage for more clarity in introduction (line no. 73).
    • Thank you for your suggestion. This has been revised as suggested (lines 74-81; highlighted in red)
  3. Elaborate clearly the difference between treatments that directly target the brain and nervous system (“CNS-directed”) and those that only work on the rest of the body (“systemic-only”).
    • Thank you for your suggestion. This has been clarified in the introduction (lines 91-96, highlighted in red) and in the paragraph below Table 1 (lines 340-344). We chose not to elaborate on specific CNS-penetrant agents as our review question was to recommend when CNS-penetrating therapies should be considered based on anatomical relation to the blood-brain barrier.   
  4. Point out more clearly which statements are backed by research evidence, and which are based mainly on expert opinion.
    • A paragraph below Table 1 (lines 340-344; highlighted in red) has been added to clarify these recommendations are expert opinion level. 
  5. Expand the section on future directions to include the role of new imaging methods, genetic/molecular testing, and large patient registry studies.
    • Thank you for this suggestion. We have added a paragraph to address this (lines 331-335; highlighted in red). 
  6. Include the treatment approaches in Figure 1.
    • We thank you for your suggestion. We have elected to keep Figure 1 as a reference for anatomical locations of these areas as we feel the treatment approaches are more clearly laid out in Table 1 for each area.

Reviewer 3 Report

Comments and Suggestions for Authors

Dear authors, the manuscript focuses on an important issue that still does not find unanimous consensus worldwide. Your effort to provide some reccomandations on diagnostic and therapuetic approaches is admirable in light of the difficulty of making an appropriate choice based on scarce data in the literature. 

some minor issues deserve to be comment and review:

  • In the introduction, the definition of SCNSL is incorrect. Lymphoma involvement of the CNS may be concomitant with extra-CNS sites or isolated in cases of recurrence. Please correct.
  • in the introduction, some concepts result to be redundant, please review and semplify them 

kind regards.

Author Response

  1. In the introduction, the definition of SCNSL is incorrect. Lymphoma involvement of the CNS may be concomitant with extra-CNS sites or isolated in cases of recurrence
    • Thank for allowing us to clarify this statement. We have clarified this statement (lines 44-45; highlighted in red).
  2. Please correct in the introduction, some concepts result to be redundant, please review and simplify them.
    • We have addressed redundancies in the manuscript